# Remedial colon hydrotherapy device enema as a salvage strategy for inadequate bowel preparation for colonoscopy: A retrospective cohort study

**Dongxuan Zhang**[1◉*], **Chunxia Zhao**[2◉], **Yuan Tao**[1], **Jiao Zhang**[1], **Qisheng Zhang**[1], **Da Li**[1], **Ping Ye**[1], **Xiaobo Yu**[1], **Chao Chen**[1]

1 Department of Gastroenterology, Beijing Changping Hospital of Traditional Chinese Medicine, Beijing, China, 2 Department of Neurology, Aerospace Center Hospital, Beijing, China.

◉ These authors contributed equally to this work.
* dongxuanzhang3087@163.com

## Abstract

### Objective

Colon hydrotherapy devices serve as a physiotherapeutic modality to manage colonic disorders by promoting intestinal peristalsis and enhancing gastrointestinal functionality. This study aims to assess and compare the effectiveness, safety, and tolerability of two remedial strategies for inadequate bowel preparation: colon hydrotherapy device enema and oral polyethylene glycol electrolyte powder.

### Methods

A retrospective analysis was performed on 109 patients who failed to adequately prepare for colonoscopy. These patients received remedial bowel preparation on the same day as their procedure, with 55 undergoing colon hydrotherapy enema and 54 receiving oral polyethylene glycol electrolyte powder. Patient satisfaction and tolerance were evaluated through telephone follow-up. Key metrics included the Boston Bowel Preparation Scale scores, preparation time, incidence of adverse reactions, and patient tolerance and satisfaction.

### Results

No significant differences were observed in baseline characteristics between groups (P > 0.05). The Boston Bowel Preparation Scale scores for the entire colon were 7 (3) in the hydrotherapy group and 6.5 (1) in the oral group (z = -2.075, P = 0.038). Notably, scores for the left colon were significantly higher in the hydrotherapy group [3 (1) vs. 2 (0), z = -5.586, P < 0.001]. The hydrotherapy group also exhibited a shorter preparation time [80 (20) min vs. 92.5 (20) min, z = -3.961, P < 0.001] and a lower incidence of adverse effects (36.4% vs. 88.9%, $\chi^2$ = 32.035, P < 0.001). Patient satisfaction metrics, including re-selection rates and tolerance of side effects, were significantly higher in the hydrotherapy group.

**Data availability statement:** All relevant data are within the paper and its Supporting Information files.

**Funding:** The author(s) received no specific funding for this work.

**Competing interests:** The authors have declared that no competing interests exist.

## Conclusions

The colon hydrotherapy device enema is an effective, efficient, and well-tolerated method for bowel cleansing, demonstrating a low incidence of adverse events. It is recommended as an effective and safe remedial therapy for patients with inadequate bowel preparation prior to colonoscopy.

## Introduction

Colonoscopy is an essential tool for the screening, diagnosis, and treating colonic lesions. Inadequate bowel preparation can result in prolonged operating time, increased difficulty during the procedure, incomplete examination, a higher risk of missed lesions, and an elevated risk of complications. Numerous studies have identified various factors that may affect the effectiveness of bowel preparation, including constipation, high body mass index (BMI), male gender, advanced age, a history of colon surgery, and other underlying medical conditions such as diabetes mellitus, Parkinson's disease, stroke, or spinal cord injury, as well as the use of medications like antidepressants [1,2]. Given the evolving socio-economic structure, the growing elderly population, and the rising popularity of colonoscopy, the proportion of patients with risk factors for inadequate bowel preparation is increasing. With the aging population growing, individuals increasingly suffer from multiple chronic diseases simultaneously, necessitating the use of various medications for treatment. Some of these medications can induce adverse reactions in the digestive tract, leading to symptoms such as nausea and vomiting and alterations in intestinal transit. Common drugs that can cause such reactions include calcium antagonists, non-steroidal anti-inflammatory drugs (NSAIDs), anxiolytics, antidepressants, diuretics, statins, metformin, and glycosidase inhibitors [3–7]. It is therefore crucial to emphasize the assessment of inadequate bowel preparation and the development of effective and safe remedial measures, while simultaneously enhancing routine bowel preparation programme. Currently, neither domestic nor international guidelines explicitly outline the optimal remedial measures for inadequate bowel preparation without canceling the same-day examination. In clinical practice, an additional 2 liters of low-dose polyethylene glycol (PEG) electrolyte powder is frequently used as a remedial approach. Several clinical studies suggest that utilizing additional oral PEG electrolytes and routine cleansing enemas may enhance bowel cleansing [8–11]. However, the quality of the existing literature is low, and comprehensive comparisons between different regimens are lacking.

The colon hydrotherapy device is primarily used in clinical settings to treat colon-related disorders such as functional constipation, colonic drug dialysis, chronic colitis, and bowel cleansing. In clinical practice, it is also employed for remedial bowel preparation. In certain cases, colon hydrotherapy devices have shown efficacy, efficiency, and safety in this role. However, there is a lack of systematic and standardized clinical studies examining the impact of remedial enemas using colon hydrotherapy devices on bowel cleanliness. The study is a retrospective cohort analysis comparing the effect of two different remedies: a colon hydrotherapy device enema and re-administration of oral polyethylene glycol (PEG) electrolyte powder. The study focuses on the improvement on colon cleanliness, the time required for remedial consumption, and patient tolerance and satisfaction among those without canceling the same-day examination.

## Materials and methods

### General information

At our endoscopy center, the standard bowel preparation protocol prior to colonoscopy involves the administration of 3 liters of polyethylene glycol (PEG). This protocol is divided into a

single-dose or a split-dose regimen. Based on both national and international literature, as well as the characteristics of our population, risk factors for inadequate bowel preparation in China include chronic constipation, a body mass index (BMI) greater than 25 kg/m², age over 70 years, a history of colon surgery, and the presence of comorbidities such as diabetes mellitus, Parkinson's disease, stroke, or spinal cord injury. The use of tricyclic antidepressants or narcotic analgesics, such as opioids, also increases this associated risks. For patients with these risk factors, we recommend administering 1 liter of PEG 6 to 10 hours before the examination and an additional 2 liters 4 to 6 hours before the procedure on the same day. For patients without clear risk factors, we advise the consumption of 3 liters of PEG 4 to 6 hours before the examination on the day of the procedure. In our daily clinical practice at the Gastroenterology Department and Endoscopy Centre, we conduct an initial assessment of bowel cleanliness for patients scheduled for colonoscopy on the day of their examination. This evaluation is based on the patient's most recent bowel movement, and the findings are documented in the patient's medical record. The assessment is categorized into four grades: Grade 1 indicated very poor bowel preparation with brown fecal water and a large amount of waste material; Grade 2 indicated relatively poor preparation with dark yellow fecal water and a small amount of waste; Grade 3 was considered good, characterized by yellowish clear watery stool; and Grade 4 was very good, defined by colorless clear watery stool. Grades 1 and 2 are considered indicative of inadequate bowel preparation [12]. For patients assessed with Grades 1 and 2, we inform them of the potential adverse effects of insufficient bowel preparation and suggest remedial measures while ultimately respecting the patient's decision.

The study included 121 patients who exhibited inadequate bowel preparation following routine administration of either a single or divided oral intake of polyethylene glycol electrolyte solution for colonoscopy at Beijing Changping Hospital of Traditional Chinese Medicine from April 2019 to June 2022. Among these, 3 patients did not complete the colonoscopy due to scope access difficulties or intolerable abdominal pain, and 9 patients declined to participate in telephone follow-ups. Consequently, 109 patients were included in the final analysis. Of these, 55 patients received treatment with an enema utilizing a colon hydrotherapy machine (DJS-C, Hangzhou Hercules Medical Instrument Co., Ltd., in China), while 54 patients underment re-administration of the oral polyethylene glycol electrolyte solution (Fig 1). The clinical case information was obtained from the medical records (S1 File). The clinical data were collected retrospectively between October 2022 and December 2023. Telephone follow-up interviews (S2 File) were also conducted from October 2022 to December 2023, with verbal consent obtained from patients in compliance with ethical standards. The study was approved by the Medical Ethics Committee of Changping Hospital of Traditional Chinese Medicine (ethics approval number: 2022-04-01).

## Inclusion criteria

1. Patient bowel preparation prior to colonoscopy was deemed inadequate. The degree of intestinal cleansing before the remedial bowel preparation was evaluated based on the characteristics of the last bowel movement before the intended colonoscopy. This assessment was categorized into four grades. Grades 1 and 2 were classified as inadequate bowel preparation.

2. The patients were required to undergo remedial bowel preparation on the day of their scheduled colonoscopy, utilizing either a colon hydrotherapy device enema or consuming 2 liters of polyethylene glycol electrolyte solution.

3. Patients who consent to and complete a telephone follow-up are eligible for inclusion in the study.

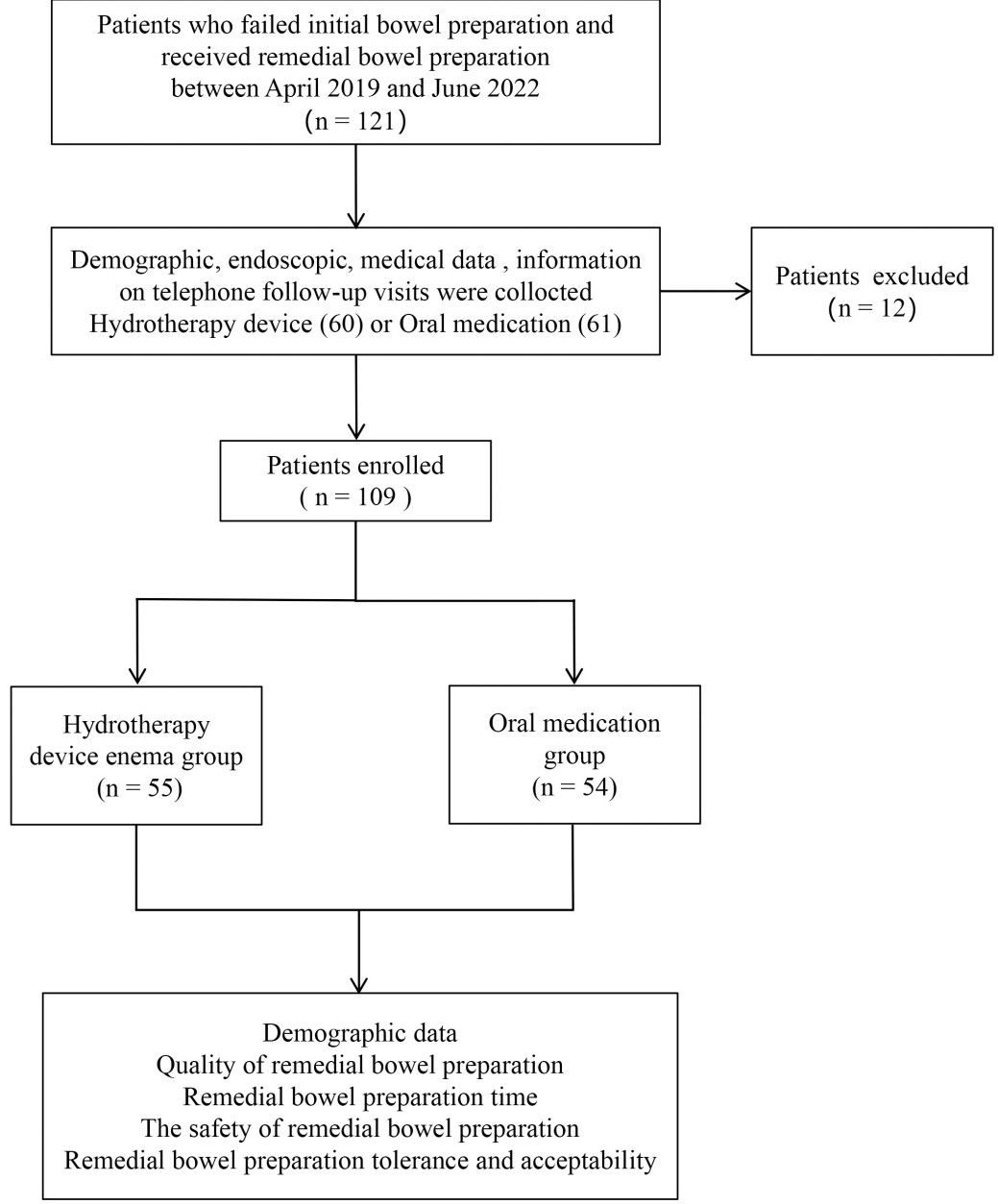

**Fig 1. The flow chart.**

## Exclusion criteria

1. Patients who used two or more remedial options in combination (different combinations of the three remedial bowel preparation methods: colon hydrotherapy device enema, oral polyethylene glycol electrolyte powder, and plain cleansing enema).

2. Patients with missing or incomplete case information.

3. Patients with incomplete telephone follow-ups.

## Remedial measures

For patients assessed as having inadequate bowel preparation after the routine bowel preparation program, remedial bowel preparation was implemented via colon hydrotherapy enema or a low-dose oral PEG program (2 L) on the same day without cancelling the examination.

The colon hydrotherapy enema procedures were conducted in the enema room (S3 File). Before commencing the procedure, patients was informed about the general operation of the colon hydrotherapy instrument. Understanding and cooperation were obtained from each patient. The patients were instructed to lie on the left side. Following the standard operating procedures of colon hydrotherapy, after necessary adjustments such as water temperature and flow rate were made and pipeline connections completed, the rectal catheter was inserted into the anus under pressure monitoring. The process of flushing and discharge continued until the colon was irrigated with the recycled liquid, resulting in either watery stools or yellowish transparent liquid.

Patients in the oral medication group were administered a low-dose PEG solution (2 L). This compound preparation consists of two doses: Dose A and Dose B. Dose A contains 13.125 grams of polyethylene glycol 4000, while Dose B contains 0.1785 grams of sodium bicarbonate, 0.3507 grams of sodium chloride, and 0.0466 grams of potassium chloride. One packet of each dose (A and B) was dissolved in 125 ml of warm water to form a solution. Patients were instructed to consume 250 milliliters every ten minutes. Once patients had passed a further number of stools, the colonoscopy procedure was initiated anew.

## Collection indicators

1. Demographic characteristics and baseline information include: gender, age, body mass index (BMI), underlying diseases, history of pelvic and abdominal surgery, Activities of Daily Living (ADL) scores, renal function evaluation, medication history, the initial bowel preparation protocol, initial bowel preparation evaluation before remedial bowel preparation.

2. Quality of remedial bowel preparation: The Boston Bowel Preparation Scale (BBPS) for the whole colon and each segment of the colon (S4 File). The BBPS is a standardized 9-point scale for assessing the colon, which is divided into three segments: right colon (cecum and ascending colon segment), transverse colon (transverse colon), and left colon (descending, sigmoid, and rectal segments). Each segment is rated from 0 to 3 based on the degree of cleanliness. The total score, which ranges from 0 to 9, indicates the level of bowel preparation: ≤ 5 points indicates poor preparation, 6–7 points indicates good preparation, and ≥ 8 points indicates very good preparation. A BBPS score of ≥ 6 was considered satisfactory. The success rate was defined as the proportion of patients in each group achieving a BBPS score of ≥ 6.

3. Remedial bowel preparation time: This refers to the interval between the initiation of remedial bowel cleansing measures and the start of the post-remedial colonoscopy. After stool assessment, patients with inadequate bowel preparation were given additional preparation, contingent upon their consent. The timing assessment was primarily based on medical records and time data from the endoscopy information system. During data collection and analysis, we extracted information from patients' medical records and recorded the start time of the post-remedial colonoscopy to evaluate the timing of remedial bowel preparation.

4. The safety of remedial bowel preparation involves documenting any adverse reactions experienced by the patient, such as nausea, vomiting, abdominal distension, abdominal

pain, dizziness, or anal itching, as noted in the medical records and reported during telephone follow-up visits.

5. Remedial bowel preparation tolerance and acceptability were evaluated by contacting patients via telephone to determine their willingness to undergo the same bowel preparation procedure for a future colonoscopy if needed. The Likert scale was used during the telephone follow-up to assess patient satisfaction with the bowel preparation process. It examined three areas: (a) satisfaction with the duration of the bowel preparation; (b) tolerance of discomfort during the procedure; and (c) satisfaction with the service attitude of the medical staff. Satisfaction was rated on a numerical scale from 0 (very dissatisfied) to 10 (very satisfied) (S2 File).

## Statistical methods

The data were analyzed using SPSS 25.0 statistical software. For data conforming to a normal distribution, results were presented as the mean ± standard deviation, and an independent samples t-test was utilized for intergroup comparisons. Non-normally distributed data were reported as the median (M) and interquartile range (IQR), with a rank-sum test applied for intergroup comparisons. Categorical data were summarized as the number of cases and corresponding percentages. The chi-square, Fisher's exact test and rank-sum test were used for group comparisons, with a P-value of less than 0.05 serving as the threshold for statistical significance.

## Results

### Comparison of general information of patients in both groups

A total of 109 patients with inadequate bowel preparation following routine administration of polyethylene glycol electrolyte solution, either as a single or divided oral intake, were included in the final analysis. Of these, 55 patients were treated with an enema using a colon hydrotherapy machine, while 54 patients received a re-administration of the oral polyethylene glycol electrolyte solution. Evaluating the general characteristics of patients in both groups was crucial for assessing comparability. The analyzed demographic and clinical features included age, gender distribution, body mass index (BMI), Activities of Daily Living (ADL) scores, prevalent underlying diseases, history of pelvic and abdominal surgeries (specifically colorectal, gallbladder, uterine and adnexal, and appendix surgeries), medications, serum creatinine levels, endogenous creatinine clearance (Ccr), the initial bowel preparation protocol and bowel preparation evaluation before remedial bowel preparation. No statistically significant differences were observed between the two groups concerning these characteristics (Table 1).

### Comparison of bowel cleansing quality following remedial bowel preparation between two patient groups

The median total Boston Bowel Preparation Scale (BBPS) score was 7 (3) in the hydrotherapy device group, compared to 6.5 (1) in the oral group, indicating a statistically significant difference in score distribution [median difference (95% CI): 1 (0-1.0), z = -2.075, P = 0.038]. In the left colon, the hydrotherapy device group achieved a BBPS score of 3 (1), which was statistically higher than the oral group, which scored 2 (0) [median difference (95% CI): 1 (0–1), z = -5.586, P < 0.001]. No statistically significant difference was found in the BBPS score distribution for the right colon, where both groups scored 2 (1), z = -0.692, P = 0.489, or for the transverse colon, where both groups scored 2 (1), z = -0.805, P = 0.421 (Table 2).

**Table 1. General characteristics of patients in both groups.**

| Variables | Total (n = 109) | Hydrotherapy device enema group (n = 55) | Oral medication group (n = 54) | Statistic | P |
|---|---|---|---|---|---|
| Age, Mean ± SD[a] | 66.29 ± 9.93 | 67.09 ± 9.669 | 65.48 ± 10.21 | $t$[b] = 0.845 | 0.400 |
| Sex, (m/[f]) | 59/50 | 27/29 | 32/22 | $\chi^2$[c] = 1.135 | 0.287 |
| BMI, Mean ± SD | 23.99 ± 3.25 | 23.89 ± 3.44 | 24.08 ± 3.08 | $t$ = -0.303 | 0.763 |
| ADL, M[d] (IQR)[e] | 95.00 (5) | 95.00 (5) | 95.00 (1) | $z$[f] = -0.298 | 0.765 |
| Diabetes, n(%) | 42 (38.5) | 25(45.5) | 17(31.5) | $\chi^2$ = 2.246 | 0.134 |
| Constipation, n(%) | 49 (45.0) | 25(45.5) | 24 (44.4) | $\chi^2$ = 0.011 | 0.916 |
| Stroke, n(%) | 29 (26.6) | 13 (23.6) | 16 (29.6) | $\chi^2$ = 0.501 | 0.479 |
| History of pelvic-abdominal surgery, n(%) | 13 (11.9) | 6 (10.9) | 7 (13.0) | $\chi^2$ = 0.109 | 0.741 |
| Colorectal surgery, n(%) | 3 (2.8) | 2(3.6) | 1(1.9) | | 1[g] |
| Gallbladder surgery, n(%) | 2(1.8) | 1(1.8) | 1(1.9) | | 1[g] |
| Uterus and adnexa, n(%) | 5(4.6) | 2(3.6) | 3(5.6) | | 0.679[g] |
| Appendix surgeries, n(%) | 5(4.6) | 1(1.8) | 4(7.4) | | 0.206[g] |
| Renal function evaluation | | | | | |
| Serum creatinine | 72.35 ± 18.97 | 70.85 ± 19.95 | 73.87 ± 17.97 | $t$ = -0.827 | 0.409 |
| Endogenous Creatinine Clearance (Ccr) | 79.55 ± 22.27 | 79.19 ± 20.98 | 79.92 ± 23.71 | $t$ = -0.172 | 0.863 |
| Medication history, n(%) | | | | | |
| Calcium antagonists | 34(31.1) | 18(32.7) | 16(29.6) | $\chi^2$ = 0.122 | 0.837 |
| NSAIDs | 22(20.2) | 8(14.5) | 14(25.9) | $\chi^2$ = 2.191 | 0.159 |
| Anxiolytic and antidepressant drugs | 10(9.2) | 6(10.9) | 4(7.4) | $\chi^2$ = 0.401 | 0.742 |
| Diuretics | 20(18.3) | 7(12.7) | 13(24.1) | $\chi^2$ = 2.342 | 0.144 |
| Statins | 24(22.0) | 11(20.0) | 13(24.1) | $\chi^2$ = 0.263 | 0.650 |
| Metformin | 17(15.6) | 8(14.5) | 9(16.7) | $\chi^2$ = 0.093 | 0.797 |
| Glycosidase inhibitors | 20(18.3) | 12(21.8) | 8(14.8) | $\chi^2$ = 0.892 | 0.459 |
| The initial bowel preparation protocol (single dose/ split dose) | 19/90 | 8/47 | 11/43 | $\chi^2$ = 0.642 | 0.423 |
| Bowel preparation evaluation(Grade 1/ Grade 2) before remedial bowel preparation | 55/54 | 26/29 | 29/25 | $\chi^2$ = 0.451 | 0.502 |

[a]Standard deviation.

[b]t-test.

[c]Chi-square test.

[d]Median.

[e]Interquartile range.

[f]Mann–Whitney test.

[g]Fisher's exact test.

## Comparison of success rates of remedial bowel preparation procedures between two patient groups

The success rate of bowel preparation was 80% in the hydrotherapy device group and 87% in the oral group. There was no statistically significant difference in the success rates between the two groups ($\chi^2$ = 0.979, P = 0.323) (Table 3).

## Comparison of remedial bowel preparation time between two groups

The median remedial bowel preparation time, with the interquartile range (IQR) was 80 [20] minutes in the hydrotherapy device group and 92.5 [20] minutes in the oral medication group.

**Table 2.  Comparison of the BBPS score after remedial bowel preparation between two patient groups.**

| | Hydrotherapy device enema group (n = 55) | Oral medication group (n = 54) | Median difference (95%CI) | Z[a] | P |
|---|---|---|---|---|---|
| Whole colon M[b] (IQR[c]) | 7(3) | 6.5(1) | 1(0-1.0) | -2.075 | 0.038 |
| The right colon M (IQR) | 2(1) | 2(1) | 0(0) | -0.692 | 0.489 |
| The transverse colon M (IQR) | 2(1) | 2(1) | 0(0) | -0.805 | 0.421 |
| The left colon M (IQR) | 3(1) | 2(0) | 1(0-1.0) | -5.586 | <0.001 |

[a] Mann–Whitney test.

[b] Median.

[c] Interquartile range.

**Table 3.  Comparison of success rates of remedial bowel preparation between the two groups.**

| Groups | Total (cases) | Pass [cases(%)] | Failed [cases(%)] | Chi-Square test | |
|---|---|---|---|---|---|
| | | | | $\chi^2$ | P |
| Hydrotherapy device enema group | 55 | 44(80.0) | 11(20.0) | 0.979 | 0.323 |
| Oral medication group | 54 | 47(87.0) | 7(13.0) | | |

There was a statistically significant difference in the distribution of remedial bowel preparation time between the two groups (median difference -15, 95% CI: -20 to -10, z = -3.961, *P* < 0.001) (Table 4).

## Comparison of adverse reaction incidence in remedial bowel preparation between two patients groups

The overall incidence of adverse reactions during remedial bowel preparation was 36.4% in the hydrotherapy device group and 88.9% in the oral group. There was a statistically significant difference in the overall incidence of adverse reactions between the two groups ($\chi^2$ = 32.035, *P* < 0.001). Specific adverse reactions, such as nausea (0% vs. 77.8%, $\chi^2$ = 69.594, *P* < 0.001), vomiting (0% vs. 31.5%, $\chi^2$ = 20.514, *P* < 0.001), abdominal distension (12.7% vs. 40.7%, $\chi^2$ = 10.95, *P* = 0.001), and abdominal pain (7.3% vs. 24.1%, $\chi^2$ = 5.843, *P* = 0.016), were significantly less frequent in the hydrotherapy group compared to the oral group. However, the incidence of dizziness (9.1% vs. 5.6%, $\chi^2$ = 0.501, *P* = 0.479) and anal itching and discomfort (14.5% vs. 3.7%, $\chi^2$ = 3.844, *P* = 0.093) did not differ significantly between the two groups (Table 5, Fig 2).

## Comparison of satisfaction and tolerance of remedial bowel preparation between two patients groups

A total of 72.7% of patients in the hydrotherapy device group indicated that they would opt for the same remedy if remedial bowel preparation was needed for another colonoscopy. In

**Table 4.  Comparison of remedial bowel preparation time between two groups.**

| Groups | M(IQR)(min) | Median difference (95%CI) | Wildcoxon two-sample rank sum test | |
|---|---|---|---|---|
| | | | Z | P |
| Hydrotherapy device enema group | 80 [20] | -15(-20- -10) | -3.961 | <0.001 |
| Oral medication group | 92.5 [20] | | | |

**Table 5. Comparison of adverse reaction incidence in remedial bowel preparation between two patient groups.**

| | Hydrotherapy device enema group (n = 55) | Oral medication group (n = 54) | $\chi^2$ | P |
|---|---|---|---|---|
| Overall Adverse Reactions n(%) | 20(36.4) | 48(88.9) | 32.035 | <0.001 |
| Nausea n(%) | 0(0) | 42(77.8) | 69.594 | <0.001 |
| Vomiting n(%) | 0(0) | 17(31.5) | 20.514 | <0.001 |
| Bloating n(%) | 7(12.7) | 22(40.7) | 10.95 | 0.001 |
| Abdominal pain n(%) | 4(7.3) | 13(24.1) | 5.843 | 0.016 |
| Dizziness n(%) | 5(9.1) | 3(5.6) | 0.501 | 0.479 |
| Anal itching n(%) | 8(14.5) | 2(3.7) | 3.844 | 0.093 |

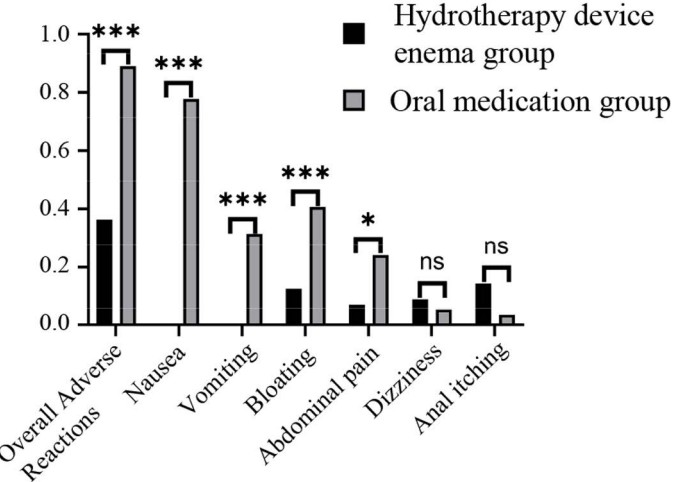

**Fig 2. Adverse reaction incidence in remedial bowel preparation between two patient groups ns, no significance; * , P ≤ 0.05; **, P ≤ 0.01; ***, P ≤ 0.001.**

contrast, 5.5% would refuse to the procedure, while 21.8% were undecided during telephone follow-up interviews. In the oral group, 31.5% expressed willingness to choose the same remedy again, 38.9% would refuse, and 31.5% were undecided. There was a statistically significant difference in the rates of choice between the two groups ($\chi^2$ = 23.345, $P$ < 0.001) (Table 6).

The median satisfaction with the time consumed for remedial bowel preparation was 9 (2) in the hydrotherapy device group and 7.5 (3) in the oral medication group. This difference was statistically significant [Median difference (95%CI) 1 (1–2), z = -5.155, $P$ < 0.001]. The median tolerance for discomfort symptoms was 10 (2) in the hydrotherapy group and 6.5 (4) in the oral group. A statistically significant difference was observed between the two groups [Median difference (95%CI) 2 (2–3), z = -5.677, $P$ < 0.001]. The median satisfaction with healthcare professionals was 10 (0) in the hydrotherapy group and 8 (4) in the oral group. The two groups were statistically different [Median difference (95%CI) 1 (0–2), z = -5.14, $P$ < 0.001] (Table 7).

The grouping of patients based on their decisions showed no significant differences in age, gender, or success rates of remedial bowel preparation among the three groups. However, significant differences were observed in the incidence of adverse reactions ($\chi^2$ = 38.177, $P$ < 0.001), Boston Bowel Preparation Scale (BBPS) scores after remedial preparation (z = 8.747,

**Table 6. Comparison of the acceptance of remedial bowel preparation in two patients groups.**

| Groups | Total (cases) | Select again[cases(%)] | Rejected[cases(%)] | Unable to select[cases(%)] | Chi-Square test | |
|---|---|---|---|---|---|---|
| | | | | | $\chi^2$ | P |
| Hydrotherapy device enema group | 55 | 40(72.7) | 3(5.5) | 12 (21.8) | 23.345 | <0.001 |
| Oral medication group | 54 | 17(31.5) | 21(38.9) | 17(31.5) | | |

**Table 7. Comparison of the satisfaction and tolerance of remedial bowel preparation in two groups of patients.**

| | Hydrotherapy device enema group (n = 55) | Oral medication group (n = 54) | Median difference (95%CI) | Z | P |
|---|---|---|---|---|---|
| Time Consumption Satisfaction M (IQR) | 9(2) | 7.5(3) | 1(1–2) | -5.155 | <0.001 |
| Tolerance of discomfort symptomsM (IQR) | 10(1) | 6.5(4) | 2(2–3) | -5.677 | <0.001 |
| Satisfaction with healthcare staff M (IQR) | 10(0) | 8(4) | 1(0-2) | -5.140 | <0.001 |

$P = 0.013$), time consumption satisfaction scores ($z = 38.78$, $P < 0.001$), tolerance of discomfort symptoms scores ($z = 75.436$, $P < 0.001$), and satisfaction with healthcare staff scores ($z = 41.915$, $P < 0.001$). The "inability to choose" group had an overall adverse reaction incidence rate of 89%, which was significantly higher than that of the "re-choice" group. Additionally, scores for time consumption satisfaction, tolerance of discomfort, and satisfaction with healthcare staff were higher in the "re-choice" group compared to both the "refusal" and "inability to choose" groups (Table 8).

## Discussion

Adequate bowel preparation is crucial for high-quality colonoscopy, as it significantly affects diagnostic accuracy and therapeutic safety during the procedure. Despite routine bowel preparation, inadequate bowel preparation remains a common clinical problem. A meta-analysis of 67 studies reported that the failure rate of bowel preparation quality after routine bowel preparation ranged from 5% to 67%, with a median of 26% [1]. Inadequate bowel preparation can lead to prolonged operation times, increased difficulty in colonoscopy, incomplete examinations, a higher risk of missed lesions, and an increased risk of complications [13-16]. Another meta-analysis of eight studies demonstrated that there was no significant difference in adenoma detection rates between moderate- and high-quality bowel preparation [15]. However, low-quality bowel preparation was associated with significantly lower adenoma detection rates. Patients must schedule their colonoscopy appointments in advance and complete blood tests and electrocardiograms to assess overall health prior to the procedure. Some

**Table 8. Comparison of characteristics between patients with three different options.**

| Variables | Select again (n = 57) | Rejected (n = 24) | Unable to select (n = 28) | Statistic | P |
|---|---|---|---|---|---|
| Age, Mean ± SDa | 66.30 ± 10.41 | 64.54 ± 9.80 | 67.79 ± 9.07 | F = 0.686 | 0.506 |
| Sex, (m/f) | 30/27 | 12/12 | 17/11 | $\chi^2 = 0.705$ | 0.739 |
| BBPS | 7(2) | 6(3)* | 6.5(2) | Z = 8.747 | 0.013 |
| The success rates of remedial bowel preparation | 87.8% | 70.8% | 85.7% | $\chi^2 = 3.403$ | 0.217 |
| Overall Adverse Reactions | 35.1% | 95.8% | 89.3% | $\chi^2 = 38.177$ | < 0.001 |
| Time Consumption Satisfaction M (IQR) | 9 (1.5) | 6(3)* | 8(1)* | Z = 38.78 | < 0.001 |
| Tolerance of discomfort symptomsM (IQR) | 10 (1) | 5(1)* | 7 (1)* | Z = 75.436 | < 0.001 |
| Satisfaction with healthcare staff M (IQR) | 10 (2) | 7 (4)* | 10 (2)* | Z = 41.915 | < 0.001 |

* Statistically different compared to the Select again group.

patients may also need to discontinue anticoagulant and antiplatelet medications, as directed by the referring physician. For inadequate bowel preparation (insufficient to detect polyps larger than 5 mm), it is recommended to either terminate and reschedule the colonoscopy or attempt additional cleansing without canceling the examination [17]. Imagine a scenario where a patient, on the day of their colonoscopy, completes the procedure after opting for remedial bowel preparation due to inadequate initial preparation. This streamlined approach helps avoid re-booking or repeating pre-colonoscopy assessments and bowel preparation, thereby reducing repeat visits and associated costs, and ensuring timely, high-quality examinations. Considering successful clinical outcomes, patient preferences and other relevant economical factors, it is vital to identify a remedial bowel preparation solution that is effective, safe, and well-tolerated for same-day administration.

Polyethylene glycol (PEG) electrolytes are the most commonly used bowel cleanser [16]. Re-administration of oral PEG electrolytes is also common as a remedial measure [18]. In a study of 131 patients, it was observed that those on a 2L oral PEG regimen achieved superior bowel preparation quality, with an adequate bowel preparation rate of 81.5% [8]. However, clinical observations indicate that if the initial bowel preparation with oral PEG electrolytes results in poor cleanliness, the incidence of adverse reactions—such as nausea, vomiting, abdominal distension, and abdominal pain—tends to be higher during remedial preparation. This leads to poor patient adherence and tolerance. Therefore, there is a need for a more efficient, safe, and well-tolerated remedial bowel preparation regimen.

Colon hydrotherapy is a physical therapy device that utilizes water impact and massage to stimulate the colon, resulting in therapeutic benefits. The device is known for its straightforward operation, safety, and reliability. It can be calibrated to meet the specific needs of the patient, allowing for adjustments in water temperature, pressure, and flow to suit diverse treatment requirements. Additionally, it features an automatic cleaning mechanism to ensure the equipment's hygiene and safety. In clinical practice, colon hydrotherapy is used to manage colon-related disorders, including functional constipation, colon dialysis, chronic colitis, bowel cleansing, and other conditions.

In this study, the Boston Bowel Preparation Scale [19–21] for the entire colon as well as the left colon, were higher in the group using the colon hydrotherapy device enema compared to the oral medication group after remedial bowel preparation. This indicated that colon cleansing was more effective in the former group, with a particularly notable improvement in the left colon cleansing. Additionally, a comparison of the time required for remedial bowel preparation showed that the colon hydrotherapy device enema group required significantly less time than the oral medication group. This suggests that colon hydrotherapy device enemas are faster and more efficient than oral medication for remedial bowel cleansing.

During the course of remedial bowel preparation, no serious complications were observed in either group. However, the overall incidence of adverse reactions in the colon hydrotherapy device enema group was significantly lower than in the oral medication group. Specifically, adverse reactions such as nausea, vomiting, abdominal pain, and abdominal distension occurred less frequently in the hydrotherapy device group. The incidence of other adverse reactions, including dizziness and anal itching, was comparable between the two groups, suggesting that colon hydrotherapy device enemas may be a safer alternative to oral medication for remedial bowel preparation.

In the oral medication group, the most frequent adverse reactions were nausea, vomiting, and bloating. In some instances, these reactions led to patients not consuming the entire bowel cleansing agent or experiencing prolonged bowel preparation times. Among those who underwent hydrotherapy, the most commonly observed adverse reactions were anal itching, bloating, and dizziness. Most patients who experienced dizziness during remedial bowel

preparation were of advanced age and presented with conditions such as carotid atherosclerosis, hypertension, cerebral infarction, and other cardiovascular and cerebrovascular diseases (S5 File). Although the incidence of dizziness was slightly higher in the hydrotherapy group compared to the oral medication group, this difference was not statistically significant. Such a reaction could potentially lead to dangerous situations, such as falls. A prospective study should be conducted to further clarify the incidence and risk factors associated with dizziness as an adverse reaction during remedial bowel preparation.

Patients in the colon hydrotherapy device group demonstrated significantly higher tolerance and satisfaction scores compared to the oral medication group. Additionally, a larger proportion of patients in the hydrotherapy group opted to use the same remedy again, indicating greater tolerance and compliance with colon hydrotherapy device enemas for remedial bowel preparation. This finding contrasts with previous studies comparing regular cleansing enemas to oral bowel cleansers [22]. Previous studies have suggested that regular cleansing enemas are less tolerated and accepted than oral cleansers. However, this study attributes the high tolerance and compliance with colon hydrotherapy to several factors:

1) The adjustable temperature and flow rate of the water stream in the colon hydrotherapy device provide greater comfort while ensuring effective cleaning.

2) The device performs repeated flushing and discharging while the patient remains in the lateral position, simplifying the procedure by eliminating the need for the patient to repeatedly get up for discharging.

3) In this study, patients initially underwent bowel preparation with oral compound polyethylene glycol electrolyte powder and were subsequently transitioned to remedial bowel preparation. It seems that the patients lacked confidence in repeating the same initial bowel preparation method. However, the different approach for remedial preparation appeared to boost their confidence and alleviate anxiety.

The endoscopy center in our hospital is currently integrated with the gastroenterology ward, which includes a dedicated enema room equipped with a hydrotherapy device. This retrospective study involved inpatients scheduled for colonoscopy due to various indications. A nurse operates the colon hydrotherapy machine daily. This machine is used for bowel cleansing and often for treating chronic constipation and administering medication retention enemas. When a colonic hydrotherapy enema is needed, the attending physician coordinates with the nurse to ensure the procedure is conducted promptly.

In this retrospective cohort study, colon hydrotherapy device enemas for remedial bowel preparation demonstrated superiority in bowel cleansing, safety, and tolerability. However, the study was limited by its non-randomised controlled trial design, small sample size, and potential recall bias. Further validation is needed through multicentre, large-sample, prospective randomised controlled trials. Additionally, as the remedial bowel cleansing program is implemented in clinical practice, further investigation into the coordination among attending physicians, the endoscopy center, nursing staff, and patients is necessary. In our healthcare system, the cost of administering 2 liters of Compound Polyethylene Glycol Electrolyte Powder orally is RMB 69.3, whereas a hydrotherapy enema costs RMB 56. Thus, hydrotherapy enema for bowel preparation is less expensive for patients compared to the oral administration of Compound Polyethylene Glycol Electrolyte Powder. Further studies will be conducted to compare the cost-effectiveness of these two methods. This will enhance the program's adaptability and facilitate its broader adoption.

## Conclusions

The proportion of patients with risk factors for inadequate bowel preparation is increasing in clinical practice. To address this issue, it is essential to continue improving the routine bowel preparation plan [23], effectively educate patients [17, 19, 23], and enhance the assessment of bowel cleanliness before and during colonoscopy [24]. Additionally, implementing efficient, safe, and well-tolerated remedial bowel preparation measures is crucial for ensuring high-quality colonoscopy examinations. In conclusion, the colon hydrotherapy device enema is a highly effective method for bowel cleansing, characterized by a short preparation time, a low incidence of adverse events, and high tolerability. Consequently, it is an effective and safe remedial therapy for patients with inadequate bowel preparation prior to colonoscopy.

## Supporting information

**S1 File. All data underlying the findings.**
(XLSX)

**S2 File. Telephone Follow-up Survey.**
(DOCX)

**S3 File. Colonic hydrotherapy device operation procedure.**
(DOCX)

**S4 File. The boston bowel preparation scale.**
(DOCX)

**S5 File. Information on 8 patients with dizziness symptoms.**
(XLSX)

## Author contributions

**Conceptualization:** Dongxuan Zhang.

**Data curation:** Dongxuan Zhang, Jiao Zhang, Qisheng Zhang.

**Formal analysis:** Dongxuan Zhang, Yuan Tao, Chao Chen.

**Investigation:** Da Li.

**Project administration:** Dongxuan Zhang.

**Resources:** Ping Ye, Xiaobo Yu.

**Software:** Qisheng Zhang.

**Writing – original draft:** Dongxuan Zhang.

**Writing – review & editing:** Chunxia Zhao.

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
