## [Decision Letter · Decision Letter 0]

14 Nov 2024

PONE-D-24-32926Remedial colon hydrotherapy device enema as a salvage strategy for inadequate bowel preparation for colonoscopy: A retrospective cohort studyPLOS ONE

Dear Dr. Zhang,

Thank you for submitting your manuscript to PLOS ONE. After careful consideration, we feel that it has merit but does not fully meet PLOS ONE’s publication criteria as it currently stands. Therefore, we invite you to submit a revised version of the manuscript that addresses the points raised during the review process.

We look forward to receiving your revised manuscript.

Kind regards,

Chih-Wei Tseng

Academic Editor

PLOS ONE

Journal Requirements:

Reviewers' comments:

Reviewer's Responses to Questions

**Comments to the Author**

1. Is the manuscript technically sound, and do the data support the conclusions?

Reviewer #1: Partly

Reviewer #2: Partly

Reviewer #3: Yes

2. Has the statistical analysis been performed appropriately and rigorously? 

Reviewer #1: Yes

Reviewer #2: Yes

Reviewer #3: I Don't Know

3. Have the authors made all data underlying the findings in their manuscript fully available?

Reviewer #1: Yes

Reviewer #2: Yes

Reviewer #3: Yes

4. Is the manuscript presented in an intelligible fashion and written in standard English?

Reviewer #1: Yes

Reviewer #2: Yes

Reviewer #3: Yes

5. Review Comments to the Author

Reviewer #1: This retrospective study aims to compare the effectiveness of hydrotherapy and oral polyethylene glycol (PEG) as alternative methods for bowel preparation in salvage cleaning for colonoscopy. There are several concerns to this paper as the following:

Major concerns:

1. Line 111: The extended time between the colonoscopy exam and the follow-up phone call may affect the patient's ability to recall details of the preparation and procedure. What were the phone call questionnaires?

2. Line 123 -130: The BBPS score or stool assessment prior to salvage cleansing should be presented.

3. Line 128-130: Is the BBPS score assessed during the first colonoscopy view, and if BBPS score <6 or <2 in any segment, is salvage cleaning then applied? Please clarify this point.

4. Line 142: The timing and method for assessing bowel cleanliness before the salvage cleaning preparation were not provided in the article.

5. Line 184, 363-364: What are the questionnaires consisted in the phone calls?? Please provide the questionnaires and chart as supplement data. How is the patient satisfaction score graded? Is the Leaker Score used?

6. Line 213: The difference between BBPS scores of 7 and 6.5 is minimal, especially given the study's small sample size. Hydrotherapy appears to be most effective in cleaning the descending colon, as shown in Table 2, but it seems to have limited impact on the ascending and transverse colons. Therefore, it is recommended to increase the sample size to enhance the reliability of the observed differences.

7. Line 229: How is the success rate defined? Was cleanliness graded and compared before and after salvage cleaning? What is the average experience in years of the colonoscopists? Please provide the actual data in chart or supplementary data.

8. It would be interesting to compare the ADR between the two groups.

9. Line 358-360: The supplement data or chart did not provide information on conditions such as carotid atherosclerosis, hypertension, cerebral infarction, and numerous other cardiovascular and cerebrovascular diseases. On what basis was this conclusion made?

10. The BBPS score shown in the S1 file assessed after applying salvage strategy?

Minor concerns:

1. English writing skills should be improved.

2. The make and model of the hydrotherapy machine should be included in the method or supplementary S2 file.

3. The labels in Table 2 are unclear. “Hepatic, transverse and splenic flexure” is confusing.

4. Line 342-343: redundant, repeated and unclear statement.

5. Reduce the use of long sentences. For example in line 394 to 399.

6. In S1 file, cases number 25, 71, and 82: The “colonoscopy” column indicates “poor bowel preparation”, yet the BBPS score is still 6. Please clarify.

6. Please review all the punctuation.

Reviewer #2: It has been my pleasure to review the manuscript titled “Remedial colon hydrotherapy device enema as a salvage strategy for inadequate bowel preparation for colonoscopy: A retrospective cohort study”. As authors say, it is a priority to guarantee a correct preparation before colonoscopy to improve the rates of diagnosis during the procedure. Preparation protocols may fail due to bad tolerance, mistakes during the performance, patients’ features, etc.; so it is necessary to have alternative possibilities. It is also very important that patients do not have poor experiences during the preparation so they agree to continue controls when needed.

Authors make with this article a very interesting proposal. However there are many points I recommend further development before accepting this manuscript for publication.

Comments about the preparation:

· There is a detailed explanation about second preparations, but not about the main protocol of colon preparation. All patients followed a standard protocol? Could there be differences?

· Was there any evaluation of previous experience in colonoscopy?

· If I understand correctly, patients with inadequate colon preparation have to do a second treatment after a colonoscopy. So after sedation, they are informed and make a choice between hydrotherapy and oral medication. How much time passes between they awake and start the second preparation?

Comments about the patients, selection criteria and potential bias:

· How many patients were excluded of the final evaluation? Criteria is explained, but there is no further information.

· It is a very important limitation the N of patients and do not allow subgroups. As we see in the S1 file, their baseline characteristics in medical antecedents are very different. For example, there are patients that have incomplete intestinal obstruction, but no information of the level is given. Further, history of abdominal surgery is a very vague term. As we can see in the file, there are included in a same group colorectal tumor surgeries, gynecological surgeries, etc. without differentiation. They are important details for colonic transit and preparation results.

· No information about baseline medications is given, and they may influence on final results of preparations.

· Hydrotherapy or oral medication were personal choices, although there are some that said they were unable to make a choice. What was the reason in these cases? Was there any evaluation of preparation failure in previous studies? This may influence on personal preferences and experience of the patient.

· Follow-up interviews were done after one year or more. Recall bias is a huge limitation here.

Comments about adverse effects:

· I understand after reading S1 file that some patients were hospitalized. Was there any evaluation of kidneys function after double preparation?

· Colon hydrotherapy may cause rectal ulcers or other lesions, even if performed by a well-experienced nurse. Were there any lesions?

Comments about the conclusions of the Authors:

· The first sentence (lines 393-394) is not well expressed. The increase of colonoscopies do not exactly increase the number of inadequate preparations. But it is true that having correct protocols and alternatives for colon preparation is a main step.

I have some suggestions about economical details to add in the Discussion Section, if it is possible, to give some more context to the article.

· What is the difference between repeating the colonoscopy the same day or another day?

· I assume there is an extra agenda in the Endoscopy Unit to repeat a patient with poor / inadequate colon preparation. How is it managed? The goal is to avoid cancellation, but as care pressure is increasing and agendas are commonly tight, it is very difficult to me to understand how it is done.

· What is the extra cost to have every day a trained nurse to perform colonic hydrotherapy, just in case it is needed?

Reviewer #3: Thank you for the opportunity to review the manuscript, "Remedial colon hydrotherapy device enema as a salvage strategy for inadequate bowel preparation for colonoscopy: A retrospective cohort study." Please find my comments below:

1. I am not familiar with the term "electronic colonoscopy" is this just a standard colonoscopy or is this a variant?

2. How did you select who received which intervention?

3. Were any of your satisfaction assessments validated tools?

4. Can you expand on how many patients were sent for remedial therapy solely based on the passage of stool just prior to their scheduled procedure vs how many had the scope placed and the BBPS evaluated? For those where the scope was placed and the BBPS was evaluated for at least 1 segment was there a difference in the change in prep quality adjustment between the 2 interventions?

5. How do you define preprocedureal constipation for the purposes of this study?

6. What is the definition for this study of "success rate" of the bowel preparation?

7. What is the difference in cost between these two measures?

6. PLOS authors have the option to publish the peer review history of their article (what does this mean? ). If published, this will include your full peer review and any attached files.

**Do you want your identity to be public for this peer review?** For information about this choice, including consent withdrawal, please see our Privacy Policy .

Reviewer #1: No

Reviewer #2: No

Reviewer #3: No

---

## [Author Response · Author response to Decision Letter 1]

26 Dec 2024

Dear Reviewers:

We are writing to express our sincere appreciation for your insightful comments and suggestions regarding our article entitled “Remedial colon hydrotherapy device enema as a salvage strategy for inadequate bowel preparation for colonoscopy: A retrospective cohort study” (Manuscript No: PONE-D-24-32926). This study explores remedial strategies to improve bowel preparation for colonoscopy, a critical aspect of effective diagnostic procedures. The constructive feedback provided by the reviewers has been invaluable in enhancing the quality of our manuscript. We have made extensive modifications and supplemented additional data to strengthen our findings and ensure the robustness of our results. All changes in the revised manuscript are highlighted in red text for your convenience. Below this letter, you will find point-by-point responses to the comments from the three reviewers. We believe these revisions have significantly improved our manuscript, and we are grateful for the opportunity to address these important issues.

Reviewer #1

Major concerns:

1. Line 111: The extended time between the colonoscopy exam and the follow-up phone call may affect the patient's ability to recall details of the preparation and procedure. What were the phone call questionnaires?

The author's answer: Thank you for your valuable comments. We apologize for not uploading the telephone questionnaire as an attachment when the manuscript was first submitted. The telephone questionnaire in this study consisted of three parts.

The first part was the Introduction, which explained the purpose and content of the telephone follow-up to the respondents and obtained their consent to proceed with the questionnaire. The second part aimed to confirm the respondent's basic personal information, including name, age, and gender. The third part consisted of seven main questions designed to gather detailed insights relevant to our study. These questions were as follows:

① Do you still clearly recall the process of your colonoscopy examination conducted at our endoscopy center in [Time]?

② Given that you underwent remedial bowel preparation for a previous colonoscopy due to inadequate preparation, would you prefer the same remedial preparation method for any future colonoscopies where bowel preparation is suboptimal? (Select again, Rejected, Unable to select)

③ On a scale from 0 to 10, where 0 is very dissatisfied and 10 is very satisfied, how would you rate the time consumed in the remedial bowel preparation process?

④ On a scale from 0 to 10, where 0 is very dissatisfied and 10 is very satisfied, how satisfied were you with the level of discomfort you experienced from the various symptoms during the remedial bowel preparation process?

⑤ On a scale from 0 to 10, where 0 is very dissatisfied and 10 is very satisfied, how satisfied were you with the service attitude of the medical staff during the remedial bowel preparation process?

⑥ Could you please detail the discomfort you encountered during your remedial bowel preparation?

⑦ Have you encountered any issues or difficulties that you would like to share?

These questions were designed to assess the respondents' experiences and satisfaction with the remedial bowel preparation process. We uploaded the telephone questionnaire as supplementary data, as S2 File. Meanwhile we have a statement in the manuscript at lines 218 to 225.

In conducting this retrospective study, we included patients who underwent remedial bowel preparation between April 2019 and June 2022 to ensure an adequate sample size for inclusion. The telephone questionnaire was conducted between October 2022 and December 2023, which introduced the potential for recall bias in the results.To minimize recall bias, our team members reviewed the patients' inpatient medical records to identify any discrepancies with the telephone questionnaire responses. This cross-verification helped ensure the accuracy of the data collected. To further address recall bias, we are planning a prospective study. This study will aim to collect information from patients immediately after they have received bowel preparation, thereby reducing the reliance on long-term memory and improving data accuracy.

2.Line 123 -130: The BBPS score or stool assessment prior to salvage cleansing should be presented.

The author's answer: We appreciate your valuable comments. In this retrospective study, the degree of bowel cleansing prior to remedial bowel preparation was assessed based on the fecal characteristics of the last bowel movement before the proposed colonoscopy. The assessment was graded into four categories:

Grade 1: Very poor bowel preparation (brown fecal water with a large amount of waste material).

Grade 2: Relatively poor (dark yellow fecal water with a small amount of fecal waste material).

Grade 3: Good (yellowish clear watery stool).

Grade 4: Very good (colorless clear watery stool).

Grades 1 and 2 were considered inadequate bowel preparation. We have revised and clarified lines 125-135 of the article to better explain this grading system. The results of the stool assessment prior to bowel preparation for each patient are provided in the S1 file.

In the manuscript, we conducted a statistical analysis of the pre-remedial bowel preparation fecal assessment results for both groups. The overall distribution was similar, with no statistical difference observed (χ² = 0.451, P = 0.502). This analysis is presented in lines 248-251 of the manuscript, and the statistical results are detailed in Table 1.

3.Line 128-130: Is the BBPS score assessed during the first colonoscopy view, and if BBPS score <6 or <2 in any segment, is salvage cleaning then applied? Please clarify this point.

The author's answer: We thank you for your helpful comments and apologize for not explaining this issue clearly in the article. In our daily clinical practice at the Gastroenterology Department and Endoscopy Centre, we perform an initial assessment of bowel cleanliness for patients undergoing colonoscopy on the day of the scheduled examination. This assessment is based on the patient's last bowel movement, and the results are recorded in the patient's medical record. For patients assessed as Grade 1 or Grade 2, we inform them of the potential adverse effects of poor bowel preparation and recommend remedial bowel preparation, while ultimately respecting the patient's choice.

In this retrospective study, we did not use the Boston Bowel Preparation Scale (BBPS) score to decide whether a patient should undergo remedial bowel preparation. During data collection, we found that only 3 of 121 patients with a left hemicolon BBPS score of less than 2 at the start of the colonoscopy were withdrawn and underwent remedial bowel preparation at the end of the procedure. The final 109 patients included in the study were judged to have failed bowel preparation based solely on their final bowel movement after routine preparation. We apologize for not detailing this in the article and are correcting and clarifying this information on lines 125-135 and lines 155-159.

4.Line 142: The timing and method for assessing bowel cleanliness before the salvage cleaning preparation were not provided in the article.

The author's answer: Thank you for pointing out this issue in the manuscript. We apologize for not explicitly stating the timing and method for assessing bowel cleanliness prior to the salvage cleaning preparation in the manuscript. We have now detailed and revised this information in lines 125-135. In our daily clinical practice at the Gastroenterology Department and Endoscopy Centre, we perform an initial assessment of bowel cleanliness for patients undergoing colonoscopy on the day of the scheduled examination. This assessment is based on the patient's last bowel movement, and the results are recorded in the patient's medical record. For patients assessed as Grade 1 or Grade 2, we inform them of the potential adverse effects of poor bowel preparation and recommend remedial bowel preparation, while ultimately respecting the patient's choice.

5.Line 184, 363-364: What are the questionnaires consisted in the phone calls? Please provide the questionnaires and chart as supplement data. How is the patient satisfaction score graded? Is the Leaker Score used?

The author's answer: We appreciate your pointing out these issues in the manuscript, and we sincerely apologize for not addressing the issues you raised in the manuscript. We uploaded the telephone questionnaire as supplementary data, as S2 File. The telephone questionnaire consisted of seven questions. The tool used for the satisfaction assessment part of the telephone follow-up was the Likert scale, which assessed patients' satisfaction with the bowel preparation process by rating the following three areas: (a) satisfaction with the time taken for the bowel preparation process; (b) tolerance of discomfort during the bowel preparation process; and (c) satisfaction with the service attitude of the medical staff during the bowel preparation process. Satisfaction was rated on a numerical scale from 0 (very dissatisfied) to 10 (very satisfied).

6.Line 213: The difference between BBPS scores of 7 and 6.5 is minimal, especially given the study's small sample size. Hydrotherapy appears to be most effective in cleaning the descending colon, as shown in Table 2, but it seems to have limited impact on the ascending and transverse colons. Therefore, it is recommended to increase the sample size to enhance the reliability of the observed differences.

The author's answer: Thank you for your valuable comments. We fully understand your concerns. The sample size of this retrospective study is indeed small, and there are unavoidable issues such as recall bias and incomplete data. To improve the reliability of the observed differences and to reduce these common issues in retrospective studies, our team is continuing to conduct relevant prospective studies. We aim to collect more subjects with complete data to support our research findings.

7.Line 229: How is the success rate defined? Was cleanliness graded and compared before and after salvage cleaning? What is the average experience in years of the colonoscopists? Please provide the actual data in chart or supplementary data.

The author's answer: Thank you for your question. We sincerely apologize for not defining "success rate" within the manuscript. In this study, the Boston Bowel Preparation Scale (BBPS) was utilized to evaluate the quality of bowel preparation in patients undergoing salvage bowel preparation. The BBPS is a standardized 9-point scale for assessing the colon, which is divided into three segments: right colon, transverse colon, and left colon. Each segment is rated from 0 to 3 based on the degree of cleanliness. The total score, which ranges from 0 to 9, indicates the level of bowel preparation: ≤5 points indicates poor preparation, 6–7 points indicates good preparation, and ≥8 points indicates very good preparation. A BBPS score of ≥6 was considered satisfactory. The success rate was defined as the proportion of patients in each group achieving a BBPS score of ≥6. Additionally, differences in BBPS scores across bowel segments and overall scores were compared between the two groups. The definition of success rate has been added to the manuscript at lines 204-205. To facilitate understanding of the scale content, we have uploaded the BBPS as supplementary data, designated as S4 File.

In this retrospective study, BBPS scores for patients were not recorded before remedial bowel preparation. the degree of bowel cleansing prior to remedial bowel preparation was assessed based on the fecal characteristics of the last bowel movement before the proposed colonoscopy. The assessment was graded into four categories:

Grade 1: Very poor bowel preparation (brown fecal water with a large amount of waste material).

Grade 2: Relatively poor (dark yellow fecal water with a small amount of fecal waste material).

Grade 3: Good (yellowish clear watery stool).

Grade 4: Very good (colorless clear watery stool).

Grades 1 and 2 were considered indicative of inadequate bowel preparation. For patients assessed with these grades, we inform them of the potential adverse effects of insufficient bowel preparation and recommend remedial measures, while ultimately respecting the patient's decision. The results of the stool assessment prior to bowel preparation for each patient are provided in the S1 file. In the manuscript, we conducted a statistical analysis of the pre-remedial bowel preparation fecal assessment results for both groups. The overall distribution was similar, with no statistical difference observed (χ² = 0.451, P = 0.502). This analysis is presented in lines 249-251 of the manuscript, and the statistical results are detailed in Table 1.

Our endoscopy center currently employs five endoscopists. One has 20 years of experience, while the remaining four have 7, 8, 10, and 4 years of experience, respectively.

8.It would be interesting to compare the ADR between the two groups.

The author's answer: Thank you for your valuable comments. We have compared the ADR between the two groups in the results section (line 291-303). The statistics are presented in Table 5, along with Fig 2, which provides a more visual comparison of the ADR between the two groups.

Fig 2. Adverse reaction incidence in remedial bowel preparation between two patient groups ns, no significance; *, P ≤ 0.05; **, P ≤ 0.01; ***, P ≤ 0.001

9.Line 358-360: The supplement data or chart did not provide information on conditions such as carotid atherosclerosis, hypertension, cerebral infarction, and numerous other cardiovascular and cerebrovascular diseases. On what basis was this conclusion made?

The author's answer: Thank you for your valuable comments. In this retrospective study, we observed that 5 patients in the hydropathy enema group and 3 in the oral medication group experienced dizziness. Such reactions could lead to dangerous situations like falls, prompting further analysis of these patients' case records to identify common characteristics. We found that the average age of these 8 patients was 74.25±5.6 years, and all had a history of cardiovascular and cerebrovascular diseases. Therefore, we reported in the article, "The majority of patients who experienced dizziness during remedial bowel preparation were of advanced age and presented with a combination of carotid atherosclerosis, hypertension, cerebral infarction, and several other cardiovascular and cerebrovascular diseases." The clinical diagnoses of all patients are detailed in S1 file, and we compiled the clinical information of these 8 patients into S5 file. We will continue prospective studies to clarify the incidence and risk factors associated with dizziness as an adverse reaction during remedial bowel preparation.

10.The BBPS score shown in the S1 file assessed after applying salvage strategy?

The author's answer: Thank you for your question. Yes. After remedial bowel preparation, the patient's bowel cleanliness was assessed using the Boston Bowel Preparation Scale (BBPS).

Minor concerns:

1. English writing skills should be improved.

The author's answer: Firstly, I would like to express my sincere gratitude for the feedback you have provided. We value your comments on improving our English writing skills. We recognize that there are deficiencies in the English expression in this submission, which may have impacted the clarity and professionalism of the paper. We sincerely apologize for this and promise to take steps to improve. To enhance the quality of the paper, we have engaged a professional English editor to conduct a thorough language review of the entire text. Additionally, we plan to use advanced language correction software to aid future writing efforts, ensuring that submitted papers meet the highest linguistic standards. And here we did not list the changes but marked in red in the revised paper. We are committed to continuously improving our English writing skills in future research and writing, ensuring that our work meets the standards of the journal.

Thank you again for your feedback and support.

2.The make and model of the hydrotherapy machine should be included in the meth

---

## [Decision Letter · Decision Letter 1]

12 Jan 2025

PONE-D-24-32926R1Remedial colon hydrotherapy device enema as a salvage strategy for inadequate bowel preparation for colonoscopy: A retrospective cohort studyPLOS ONE

Dear Dr. Zhang,

Thank you for submitting your manuscript to PLOS ONE. After careful consideration, we feel that it has merit but does not fully meet PLOS ONE’s publication criteria as it currently stands. Therefore, we invite you to submit a revised version of the manuscript that addresses the points raised during the review process.

We look forward to receiving your revised manuscript.

Kind regards,

Chih-Wei Tseng

Academic Editor

PLOS ONE

Journal Requirements:

Reviewers' comments:

Reviewer's Responses to Questions

**Comments to the Author**

1. If the authors have adequately addressed your comments raised in a previous round of review and you feel that this manuscript is now acceptable for publication, you may indicate that here to bypass the “Comments to the Author” section, enter your conflict of interest statement in the “Confidential to Editor” section, and submit your "Accept" recommendation.

Reviewer #1: All comments have been addressed

Reviewer #2: All comments have been addressed

Reviewer #3: (No Response)

2. Is the manuscript technically sound, and do the data support the conclusions?

Reviewer #1: Yes

Reviewer #2: Yes

Reviewer #3: Yes

3. Has the statistical analysis been performed appropriately and rigorously? 

Reviewer #1: Yes

Reviewer #2: Yes

Reviewer #3: Yes

4. Have the authors made all data underlying the findings in their manuscript fully available?

Reviewer #1: Yes

Reviewer #2: Yes

Reviewer #3: Yes

5. Is the manuscript presented in an intelligible fashion and written in standard English?

Reviewer #1: Yes

Reviewer #2: Yes

Reviewer #3: Yes

6. Review Comments to the Author

Reviewer #1: (No Response)

Reviewer #2: Thank for allowing to review this manuscript. The Authors have gone through all points expressed by the Reviewers very carefully.

I have some minor comments to give:

· Lines 82-83: these medications can induce secondary effects in the digestive tract, such as nausea, vomiting and (add) “alterations in intestinal transit”4.

· Lines 118-119: I am confused with the sentence “The use of tryciclic antidepressants or anesthesia also increases this risk”. Is “anesthesia” referring to analgesia / pain killers?

· Line 182: I would separate in different paragraphs these sentences: one paragraph for hydrotherapy, and one for oral medication.

· Line 446: I would separate in different paragraphs these sentences: one paragraph for the hospital organization, one for comments about the study.

· In my opinion, the conclusion given is too aggressive. The results of this retrospective study lead to think that colon hydrotherapy is a good and safe remedial therapy for patients with inadequate bowel preparation, but there is no strength in data to say that “it is recommended as the optimal remedial strategy”. I would be more cautious with this sentence.

Reviewer #3: Thank you for the opportunity to review the revised manuscript, "Remedial colon hydrotherapy device enema as a salvage strategy for inadequate bowel preparation for colonoscopy: A retrospective cohort study." Please find my comments below:

1. You list as exclusion criteria "Patients who used two or more remedial options in combination." Could you be more clear here. I agree that if they used both techniques without an evaluation between each technique it would be appropriate to exclude. However, if they used one method and had an inadequate prep necessitating use of the other technique, then they should be included as a treatment failure.

7. PLOS authors have the option to publish the peer review history of their article (what does this mean? ). If published, this will include your full peer review and any attached files.

**Do you want your identity to be public for this peer review?** For information about this choice, including consent withdrawal, please see our Privacy Policy .

Reviewer #1: No

Reviewer #2: No

Reviewer #3: No

---

## [Author Response · Author response to Decision Letter 2]

17 Jan 2025

Dear Editors and Reviewers,

We would like to express our sincere gratitude to the editors and reviewers for their thorough examination of our manuscript and for providing valuable and constructive feedback. We deeply appreciate the time and effort invested in reviewing our work, and we are confident their insights will substantially enhance the quality of our study.

Our article, titled "Remedial Colon Hydrotherapy Device Enema as a Salvage Strategy for Inadequate Bowel Preparation for Colonoscopy: A Retrospective Cohort Study" (Manuscript No: PONE-D-24-32926), explores strategies to improve bowel preparation—a crucial component of effective diagnostic procedures for colonoscopy. We are pleased that the reviewers have acknowledged the significance and novelty of our research. Their suggestions have been instrumental in refining and clarifying our findings. We have carefully considered each of their comments and made the necessary revisions to address the issues raised. For your convenience, all changes in the revised manuscript are highlighted in red text. We have rechecked the reference list of the manuscript to ensure its completeness and accuracy. The 13th and 22nd references in the article are in Chinese, and we have labeled them in the reference list according to ICMJE recommendations（line 520 and line 551）.

Below this letter, you will find point-by-point responses to the reviewers' comments.

We believe these revisions have significantly strengthened our manuscript, and we appreciate the opportunity to address these important concerns.

Reviewer #1: (No Response)

Reviewer #2: Thank for allowing to review this manuscript. The Authors have gone through all points expressed by the Reviewers very carefully.

I have some minor comments to give:

· Lines 82-83: these medications can induce secondary effects in the digestive tract, such as nausea, vomiting and (add) “alterations in intestinal transit”.

The author's answer: Thank you for your valuable comments. Your revision has made this statement more comprehensive and precise. We have incorporated this change into the manuscript (lines 83-84).

· Lines 118-119: I am confused with the sentence “The use of tryciclic antidepressants or anesthesia also increases this risk”. Is “anesthesia” referring to analgesia / pain killers?

The author's answer: We thank you for your helpful comments and apologize for not explaining this issue clearly in the article. In the manuscript, “anesthesia” is referring to narcotic analgesics, such as opioids. For greater clarity and to avoid misunderstandings, we have revised this line in the manuscript to read: "The use of tricyclic antidepressants or narcotic analgesics, such as opioids, also increases the associated risks." (lines 118-119)

· Line 182: I would separate in different paragraphs these sentences: one paragraph for hydrotherapy, and one for oral medication.

The author's answer: Thank you for your valuable advice. This revision enhances the article's structure. We have reorganized this section of the manuscript into two paragraphs: one focused on hydrotherapy and the other on oral medication. (lines 178-195)

· Line 446: I would separate in different paragraphs these sentences: one paragraph for the hospital organization, one for comments about the study.

The author's answer: Thanks again for your valuable comment. We have reorganized this section of the manuscript into two paragraphs: one focused on the hospital organization and the other on comments about the study. (lines 446-467)

· In my opinion, the conclusion given is too aggressive. The results of this retrospective study lead to think that colon hydrotherapy is a good and safe remedial therapy for patients with inadequate bowel preparation, but there is no strength in data to say that “it is recommended as the optimal remedial strategy”. I would be more cautious with this sentence.

The author's answer: Thank you for your valuable comments. We fully understand your concerns. Indeed, in this retrospective study, the conclusions were initially expressed too assertively. Based on the current study results, we recognize the need to conduct further prospective studies to validate our conclusions. Following your comments, we have revised the abstract and conclusion sections of the manuscript, which now state: "It is recommended as an effective and safe remedial therapy for patients with inadequate bowel preparation prior to colonoscopy." (lines 64-65) and "Consequently, it is an effective and safe remedial therapy for patients with inadequate bowel preparation prior to colonoscopy." (lines 477-478).

Reviewer #3: Thank you for the opportunity to review the revised manuscript, "Remedial colon hydrotherapy device enema as a salvage strategy for inadequate bowel preparation for colonoscopy: A retrospective cohort study." Please find my comments below:

1. You list as exclusion criteria "Patients who used two or more remedial options in combination." Could you be more clear here. I agree that if they used both techniques without an evaluation between each technique it would be appropriate to exclude. However, if they used one method and had an inadequate prep necessitating use of the other technique, then they should be included as a treatment failure.

The author's answer: Thank you for your valuable comments, and we apologize for not clearly presenting this in the manuscript. In this retrospective clinical study, we included "patients with a combination of two or more remedies" as an exclusion criterion. In clinical practice, the specifics of combining two or more therapeutic regimens include the following scenarios:

1.After routine bowel preparation, if a patient is initially assessed as having inadequate bowel preparation, the treating clinician may first administer a low dose of polyethylene glycol (PEG) electrolyte powder orally. If the patient is unable to continue oral intake due to severe nausea, vomiting, or other adverse reactions, or if there is no bowel movement post-intake, the clinician may communicate with the patient to proceed with either a plain cleansing enema or a colon hydrotherapy device enema to continue the bowel preparation.

2.Patients evaluated as having poor bowel preparation via fecal assessment, who did not respond to a plain cleansing enema, might receive a colon hydrotherapy device enema.

These patients undergo colonoscopy following a complex combination of bowel preparation methods, with cleanliness assessed using the BBPS scale. We determined that it was challenging to ascertain whether the bowel cleanliness was due to a specific bowel preparation regimen or the combined effect of different regimens. Therefore, we excluded patients who used combinations of two or more bowel cleansing regimens in this retrospective clinical trial. This explanation is provided in lines 167-170 of the manuscript.

Thank you again for your suggestion, which has highlighted the need to further investigate and validate the scope of colon hydrotherapy device enema as a remedial bowel preparation method. This includes not only its use for inadequate bowel preparation following conventional regimens but also exploring its value after other remedial preparation regimens.

Once again, we thank the reviewers for their support and for their role in helping us to improve our manuscript. We look forward to the opportunity to further discuss our work with you.

Yours sincerely,

Dongxuan Zhang

17 January 2025

---

## [Decision Letter · Decision Letter 2]

4 Feb 2025

Remedial colon hydrotherapy device enema as a salvage strategy for inadequate bowel preparation for colonoscopy: A retrospective cohort study

PONE-D-24-32926R2

Dear Dr. Zhang,

We’re pleased to inform you that your manuscript has been judged scientifically suitable for publication and will be formally accepted for publication once it meets all outstanding technical requirements.

Kind regards,

Chih-Wei Tseng

Academic Editor

PLOS ONE

Additional Editor Comments (optional):

All comments have been addressed

Reviewers' comments:

Reviewer's Responses to Questions

**Comments to the Author**

1. If the authors have adequately addressed your comments raised in a previous round of review and you feel that this manuscript is now acceptable for publication, you may indicate that here to bypass the “Comments to the Author” section, enter your conflict of interest statement in the “Confidential to Editor” section, and submit your "Accept" recommendation.

Reviewer #2: All comments have been addressed

Reviewer #3: All comments have been addressed

2. Is the manuscript technically sound, and do the data support the conclusions?

Reviewer #2: Yes

Reviewer #3: Yes

3. Has the statistical analysis been performed appropriately and rigorously? 

Reviewer #2: Yes

Reviewer #3: Yes

4. Have the authors made all data underlying the findings in their manuscript fully available?

Reviewer #2: Yes

Reviewer #3: Yes

5. Is the manuscript presented in an intelligible fashion and written in standard English?

Reviewer #2: Yes

Reviewer #3: Yes

6. Review Comments to the Author

Reviewer #2: (No Response)

Reviewer #3: (No Response)

7. PLOS authors have the option to publish the peer review history of their article (what does this mean? ). If published, this will include your full peer review and any attached files.

**Do you want your identity to be public for this peer review?** For information about this choice, including consent withdrawal, please see our Privacy Policy .

Reviewer #2: No

Reviewer #3: No

---

## [Editor Report · Acceptance letter]

PONE-D-24-32926R2

PLOS ONE

Dear Dr. Zhang,

I'm pleased to inform you that your manuscript has been deemed suitable for publication in PLOS ONE. Congratulations! Your manuscript is now being handed over to our production team.

Kind regards,

on behalf of

Dr. Chih-Wei Tseng

Academic Editor

PLOS ONE